# *Balantioides coli* Fecal Excretion in Hunted Wild Cervids (*Cervus elaphus* and *Dama dama*) from Portugal

**DOI:** 10.3390/pathogens11111242

**Published:** 2022-10-27

**Authors:** João Mega, Sérgio Santos-Silva, Ana Loureiro, Josman D. Palmeira, Rita T. Torres, Antonio Rivero-Juarez, David Carmena, João Mesquita

**Affiliations:** 1ICBAS—School of Medicine and Biomedical Sciences, Porto University, 4050-313 Porto, Portugal; 2Department of Biology & CESAM, University of Aveiro, Campus de Santiago, 3810-193 Aveiro, Portugal; 3CIBER Infectious Diseases (CIBERINFEC), Health Institute Carlos III, 28029 Madrid, Spain; 4Grupo de Virología Clínica y Zoonosis, Unidad de Enfermedades Infecciosas, Instituto Maimónides de Investigación Biomédica de Córdoba, Hospital Reina Sofía, Universidad de Córdoba, 14004 Córdoba, Spain; 5Parasitology Reference and Research Laboratory, National Centre for Microbiology, 28220 Majadahonda, Spain; 6Epidemiology Research Unit (EPIUnit), Instituto de Saúde Pública da Universidade do Porto, 4050-600 Porto, Portugal; 7Laboratório Para a Investigação Integrativa e Translacional em Saúde Populacional (ITR), 4050-313 Porto, Portugal

**Keywords:** *B. coli*, wildlife, deer, Portugal, one health, transmission, genotyping, surveillance

## Abstract

*Balantioides coli* is a zoonotic enteric protozoan parasite of public veterinary health relevance and a concern in animal production and food safety. While wild cervids are recognized reservoirs for several zoonotic pathogens, little is known about the occurrence of *B. coli* in deer species, especially in Europe. To fill this gap, a total of 130 fecal samples from legally hunted red deer (*Cervus elaphus*, *n* = 95) and fallow deer (*Dama dama*, *n* = 35) were passively collected during two hunting seasons (October to February; 2018–2019 and 2019–2020) in Portugal. After assessment by PCR assay targeting the complete ITS1–5.8s-rRNA–ITS2 region and the 3’ end of the *ssu*-rRNA gene of the parasite, a prevalence of 4.2% (4/95, 95% CI: 0.2–8.3) in red deer and of 5.7% (2/35, 95% CI: 0.0–13.4) in fallow deer was found. Sequence and phylogenetic analyses allowed the identification of *B. coli* genetic variants A (in two red deer) and B (in two red deer and two fallow deer). This is the first molecular-based description of *B. coli* in European deer species, whose population have increased in density and geographical range in recent years. Continued monitoring of wild ungulates as potential vectors of parasitic infection diseases of zoonotic nature is crucial to safeguard public health and food safety.

## 1. Introduction

*Balantioides coli* is a zoonotic enteric protozoan parasite of worldwide distribution that can cause mild infection to life-threatening diseases in humans and animals [1,2]. Since 2014, *B. coli* has been considered, both by the Food and Agriculture Organization of the United Nations and the World Health Organization, an emerging pathogen and a foodborne parasite that should be included in specific control guidelines [3]. This parasite is frequently found in the gastrointestinal tract of mammals, namely domestic and production animals, such as pigs (the primary reservoir host) [4], sheep [5] and horses [6]. *Balantioides coli* is the only ciliated protozoan capable of infecting humans, albeit with low prevalence [7]. In most mammals, including humans, *B. coli* infection is asymptomatic and causes no significant damage to the gastrointestinal tract [7]. However, if the host is immunocompromised, acute or chronic clinical manifestations may appear [8].

Although rare until today [2], *B. coli* is regarded as a pathogen capable of being involved in waterborne and foodborne outbreaks [9], having been detected in raw vegetables and fruits [10], animals (such as pigs) raised for meat production [11] and in water matrices (drinking water, rivers, ponds, canals and wastewater) [12]. In addition, deaths associated with *B. coli* infections have been recorded in humans [13] and animals, such as farm horses and non-human primates in captivity [14,15]. Transmission occurs via the fecal–oral route, either indirectly by ingestion of contaminated food or water or through direct contact with infected hosts or their fecal material [2,3,4,5,6,7,8,9,10,11,12,13,14,15,16]. In humans, infection by *B. coli* is characterized by abdominal pain and diarrhea. Severe cases may also lead to bleeding and perforation of the colon [17]. In chronic cases, mild recurrent diarrhea associated with strength deficit and weight loss are common [16].

Nevertheless, little is known on the epidemiology of *B. coli* in wildlife, particularly in members of the Cervidae family. There are some reports on *B. coli* infection in spotted deer (*Axis axis*) and sambar deer (*Rusa unicolor*) in Bangladesh [18,19,20]. Of these reports, one found a prevalence of 1.6% in 127 samples of wild deer in the Bhola district [18], while the other two studies, both in zoological gardens and with a diminutive number of samples available, found infection rates ranging from 50–100% depending on the species [19,20]. Wild cervids have been increasing in geographical range and density all over Europe [21], playing a central role in the transmission of zoonotic diseases including pathogenic bacteria such as *Escherichia coli*, *Pseudomonas aeruginosa*, *Enterocytozoon bieneusi* and *Mycobacterium bovis* [22,23,24,25]. Additionally, they are widely hunted and their meat consumed, placing them in direct contact with humans [21].

For the above-mentioned reasons, it is pivotal to understand the role of wild cervids in the dynamics of several zoonotic agents, such as *B. coli*. This study aims to assess the prevalence and molecular diversity of *B. coli* in wild cervids living in Portugal.

## 2. Results

From the total 130 fecal samples examined, 4.6% (6/130, [95% Confidence Interval (CI): 1.0–8.2]) tested positive for the presence of *B. coli* by ITS-*ssu*-PCR. DNA of the parasite was found in 5.7% (2/35, [95% CI: 0.0–13.4]) of fallow deer and in 4.2% (4/95, [95% CI: 0.2–8.3]) of red deer, respectively.

The results showed a higher prevalence in samples from *D. dama*. Presence of *B. coli* was also greater in female or juvenile individuals. Regions with the highest prevalence were Coimbra and Lisbon. All samples were formed, having the same fecal consistency. Statistical analysis was completed using Chi-square (χ^2^) test, demonstrating that none of the variables considered (host species, sex, age group, geographical area of origin) were significantly associated with a higher likelihood of *B. coli* infection (Table 1). The *p*-value for fecal consistency was not calculated as all samples were formed.

Table 2 summarizes the main epidemiological features of the wild cervids that tested positive for *B. coli* in the present study.

The amplified sequences retrieved from these samples were submitted to bidirectional sequencing and subsequently confirmed as *B. coli* following BLAST analyses. Sequence similarity analysis within the six positive samples investigated in this study revealed that sequences shared 80.57–99.45% identity between them. It also indicated that sequences shared 83–99% identity with *B. coli* sequences previously deposited in the GenBank public repository isolated from pigs in South Korea (MZ676845), captive chimpanzees in Spain (JQ073346) and wild boars in China (MT258438) (Table 3).

Phylogenetic analysis showed that sequences obtained in this study fall within well-supported clusters with other *B. coli* sequences corresponding to genetic variants A and B (Figure 1). Sequences from samples C39 and C68 (both from red deer) grouped within the variant A cluster, while sequences from samples C24 and C143 (red deer) and C120 and C140 (fallow deer) grouped within the variant B cluster (Figure 1).

Sequences generated in the present study were deposited in GenBank under accession numbers OM349058–OM349060 and OM349062–OM349064.

## 3. Discussion

This molecular-based study reports for the first time the presence and genetic diversity of the ciliate enteroparasite *B. coli* in wild deer in Europe. Infection rates were 4.2% in red deer and 5.7% in fallow deer. The fact that *B. coli* was always detected in formed fecal samples is indicative of subclinical infections not linked with gastrointestinal manifestations. Although results show slight differences in prevalence in variables including host species, sex, age group and geographical area of origin, none presented statistical relevance; thus, were not associated with a higher likelihood of *B. coli* infection. Our results show that *B. coli* circulates among wild cervids in Portugal.

An early study conducted in Portugal in 2020 failed to demonstrate the presence of *B. coli* in 88 deer (73 red deer and 15 roe deer) fecal samples by microscopy examination [26]. More recently, *B. coli* was also undetected in a large PCR-based epidemiological survey at national scale conducted in neighboring Spain involving 1023 DNA fecal samples from wild cervid (fallow deer, red deer, and roe deer) and bovid (Barbary sheep, Iberian wild goat, mouflon, and Southern chamois) species [27]. Taken together, these data suggest that differences in epidemiological (infection pressure), environmental (climatic conditions, geographical areas) and host-dependent (species, age, immunological status) variables might influence the transmission of *B. coli* in the Iberian Peninsula. 

Out of Europe, three studies have identified the presence of *B. coli* in wild and captive deer species in Bangladesh [18,19,20]. One in Bhola District, that presented a prevalence of 1.6% in 127 samples, in wild deer samples [18]. Other two studies also tested deer but not from wild populations, such as our, but from managed populations in zoos (Dhaka and Rangpur Zoological Gardens). At Rangpur Zoological Garden, 23 samples of spotted deer (*Axis axis*) tested negative, while the only sample of sambar deer (*Rusa unicolor)* tested positive [19]. At Dhaka Zoological Garden, from six samples of spotted deer, half tested positive while there were three positive cases of infection from four samples of sambar deer [20]. Differences in fecal shedding prevalence could be due to a high animal density in zoos, likely favoring the transmission of enteric pathogens. However, caution should be taken as a very limited number of samples were tested at the zoological gardens. Another relevant aspect to note is that in all the studies mentioned above, identification of *B. coli* was completed by microscopic examination of fecal samples and molecular data were completely lacking [18,19,20], known to be prone to errors due to morphological characteristics being shared by cysts of other ciliate species, with sequencing as the preferred method for identification [9].

An asset of our study is that PCR-positive samples were confirmed by Sanger sequencing, allowing for accurate identification and genotyping of the obtained *B. coli* isolates. The marker of choice (the ITS1-5.8s rRNA-ITS2 gene) is particularly suited for molecular epidemiological investigations, combining high sensitivity with the possibility of differentiating the two main genetic variants (A and B) described within *B. coli* at this marker [28,29]. *Balantioides coli* variant A has been previously described in humans, pigs, gorillas and ostriches, whereas variant B has been identified in pigs, gorillas and ostriches [28,29].

After sequencing the six positive samples in our study, two sequences were found to cluster with variant A (only detected in *C. elaphus*) and four sequences with variant B (detected in both *C. elaphus* and *D. dama*) showing that genetically diverse variants co-circulate in wild deer of Portugal. However, the high variability of the marker used for the description of those variants questions the applicability of its use for analyzing the intraspecific genetic differences of *B. coli* and their taxonomic or epidemiological repercussions [30].

Little is still known about *B. coli* circulation in animals and humans. Swine are considered the main reservoir for infection [4] and, therefore, most reports on *B. coli* prevalence in farms are based on pigs. Several studies worldwide have identified high rates of infection in swine: in Denmark, most pig groups reared in a large swine farm showed fecal prevalences of 100% [31]; in Germany, 16 of 20 pig breeding farms had circa 60% infected piglets [32] and in Italy, 28–65% infection rates were found, differing for commercial hybrids pigs and autochthonous ones [4].

Although largely confined to the tropical and sub-tropical regions, cases of human infection by this parasite have been reported in other areas of the world, such as France [33] and Turkey [16]. Furthermore, in endemic regions, there have already been cases of outbreaks in humans such as in the Caroline Islands in the western Pacific Ocean [34]. Interestingly, pig farmers of Papua New Guinea were found to be occupationally exposed to *B. coli* and showed fecal excretions as high as 28% [35].

Some constraints to our study are worth mentioning as they could have some impact on results. It is often challenging to gather wildlife samples. We overcame this problem by accompanying authorized hunters during the legal gaming season and collecting samples from hunted animals, but this resulted in a smaller sample size than ideal and only two deer species (*Cervus elaphus* and *Dama dama*) ended up being represented. This smaller sample size may have affected the ability to determine statistically significant differences in *B. coli* between variables (host species, sex, age groups, and geographical area of origin). There was also a lack of quantification of cyst excretion in positive samples, which could be important in assessing the contribution of these hosts to the environmental contamination with cysts of the parasite. This task should be addressed in future studies.

Our study shows that a number of wild deer in Portugal are infected with *B. coli*, possibly having a role as reservoir hosts for infection in other animals and humans. Additionally, as deer species have been increasing in distribution and number in Portugal [36], with the same tendency in Europe [21] and have direct and indirect contact with humans through hunting activities and are widely consumed as game meat, it is essential to determine their role as potential reservoirs of *B. coli*. Just in the 2019/2020 hunting season alone, 2.240 deer were hunted in Portugal, showing an increase of 22% with previous hunting seasons (Relatório de Actividade Cinegética, 2020–2021). This highlights the importance of investigating food-borne diseases in wild animals, particularly those with in close association with humans, with continued efforts in monitoring reservoirs of *B. coli* as a crucial step to safeguard public health.

## 4. Materials and Methods

Sampling was conducted at a national scale in mainland Portugal during 2 hunting seasons (October to February; 2018/2019 and 2019/2020). Fresh stool samples (*n* = 130) were obtained directly from the rectum of hunted cervids, within 1–3 h after death. Samples were from red deer (*Cervus elaphus*; *n* = 95; 42 females and 53 males) and fallow deer (*Dama dama n* = 35; 21 females and 14 males). Red deer samples were collected from animals hunted in municipal hunted grounds, where animals roam freely as these areas are not fenced, while fallow deer samples were collected from touristic hunting grounds (some fenced, others not). Red and fallow deer densities were not available, but there were 2440 individuals hunted (all deer species) in the 2020/2021 hunting season, with 0.08 deer hunted per 100 hectares of hunting area (Relatório de Atividade Cinegética, 2020–2021). All fecal samples collected were formed. Sampling took place in three regions of Portugal, namely south (Alentejo) (*n* = 53), center (*n* = 46) and center-west (Lisbon and Tejo Valley) (*n* = 31). There was 1 sample collected both in Castelo Branco and in Setúbal, 10 in Coimbra, 23 in Portalegre, 30 both in Évora and in Lisboa and 35 in Leiria (Figure 2).

Fecal samples were frozen at −20 °C until being tested. This sampling method enables an accurate evaluation of animal health, while circumventing some of the difficulties of collecting reliable data from wild animals.

No animals were sacrificed for the purpose of this study and the authors were not responsible for any animal demise. Animals were hunted by authorized hunters during the legal gaming season, in legally organized hunting activities. Animal handling and hunting was conducted according to Portuguese National legislation.

Stools were preserved at –20 °C until processed. Fecal suspensions (10%) were prepared in phosphate-buffered saline pH 7.2 and centrifuged at 8000× *g* for 5 min. DNA extraction was performed from 140 μL of clarified supernatants employing the QIAcube^®^ automated platform (Qiagen, Hilden, Germany) and QIAamp DNA mini kit (Qiagen, Hilden, Germany). Eluted DNA was kept in RNase-free water at −80 °C.

Presence of *B. coli* was accessed by a direct PCR assay to amplify the complete ITS1–5.8s-rRNA–ITS2 region and the last 117 bp (3t’ end) of the *ssu*-rRNA sequence using the B5D (5′-GCTCCTACCGATACCGGG) and B5RC (5′-GCGGGTCATCTTACTTGATTTC) set of primers [21]. PCR reactions (25 μL) consisted of 2 μL of template DNA and 0.4 μM of each primer. PCR conditions were as follows: 95 °C for 3 min; 40 cycles of 95 °C for 30 s; 60 °C for 15 s; 72 °C for 2 s and a final extension for 10 min at 72 °C. PCR was conducted in Bio-Rad T100^TM^ Thermal Cycler and amplification products of this process were electrophoresed at 100 V for 40 min on 1.5% agarose gel stained with Xpert Green Safe DNA gel stain (Grisp, Porto, Portugal), and then irradiated with UV light to identify the target DNA fragments. A DNA weight comparison was used for measurements (100 bp DNA ladder; Grisp, Porto, Portugal).

Presumed positive amplicons were purified using GRS PCR and Gel Band Purification Kit (Grisp, Porto, Portugal). Bidirectional sequencing was performed using the Sanger Method with gene specific primers. The editing of obtained sequences was carried out using BioEdit Sequence Alignment Editor v7.1.9 software package, version 2.1 and compared with the sequences available in the NCBI (GenBank) nucleotide database (https://blast.ncbi.nlm.nih.gov/Blast.cgi (accessed on 23 October 2022)). Phylogenetic analysis was conducted using MEGA version X software [37]. Sequences identified in this study and other representative sequences, obtained from GenBank, were used for this analysis. Phylogenetic tree was drawn using the maximum-likelihood (ML) method [37,38]. The ML bootstrap values were estimated using 1000 replicates with Tamura 3-parameter model [38], which was estimated as the best substitution model by MEGA X [37]. All sequences obtained in this study were uploaded into GenBank.

## Figures and Tables

**Figure 1 pathogens-11-01242-f001:**
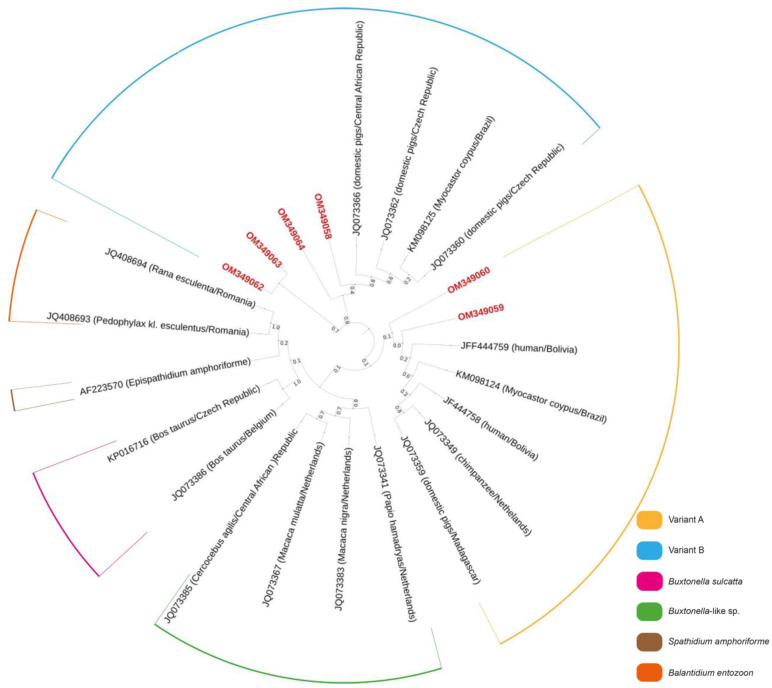
Phylogenetic tree inferred using the MEGA X maximum likelihood method (Tamura 3-parameter model) and the Interactive Tree Of Life (iTOL) based on 24 nucleotide *B. coli* sequences, including those generated this study (highlighted in bold and shaded in red). *Balantidium entozoon* is a member of the same family (*Balantidiidae*), while *Buxtonella sulcatta* (*Buxtonella*-like sp. as well) is a member of the *Pycnotrichidae*, family of the same order (*Vestibuliferida*) of *Balantidiidae*. *Spathidium amphoriforme* was used as the out-group.

**Figure 2 pathogens-11-01242-f002:**
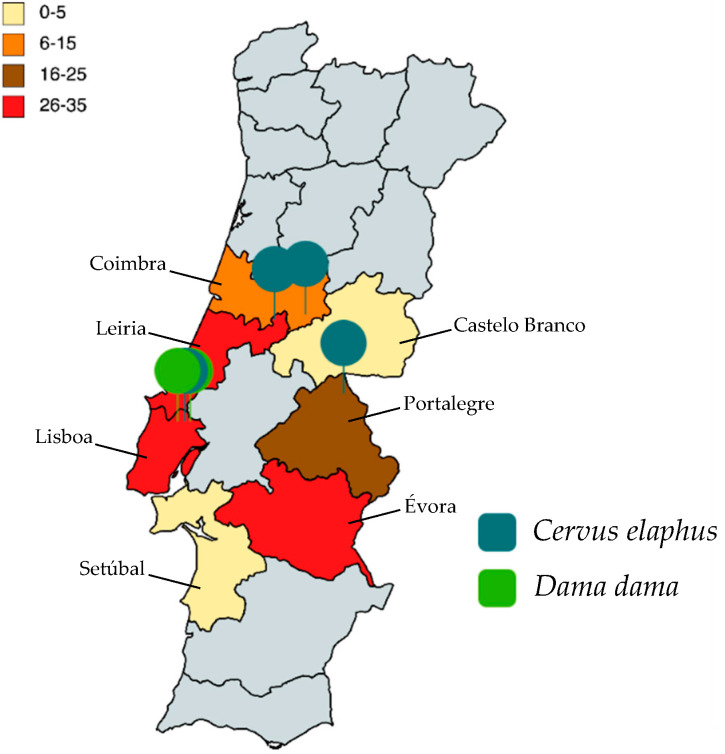
Geographic distribution of sampled deer stools. Districts in yellow are Castelo Branco and Setúbal, in orange is Coimbra, in brown is Portalegre and in red are Lisbon, Leiria and Évora. Districts are color coded according to the number of samples collected in each one. Positive cases are highlighted, with species being differentiated.

**Table 1 pathogens-11-01242-t001:** Distribution of *Balantioides coli* infections according to the epidemiological and clinical variables considered in the present study.

Variable	Total (*n*)	*B. coli* Positive (*n*)	Frequency (%)	*p*-Value
**Host species**				0.717
*C. elaphus*	95	4	4.21	
*D. dama*	35	2	5.71	
**Sex**				0.938
Male	67	3	4.48	
Female	63	3	4.76	
**Age**				0.763
Juvenile	14	1	7.14	
Semi-adult	7	0	0	
Adult	109	5	4.59	
**Fecal consistency**				---
Formed	130	6	4.62	
Diarrhea	0	0	0	
**Region**				0.718
Castelo Branco	1	0	0	
Setúbal	1	0	0	
Coimbra	10	1	10	
Portalegre	23	1	4.35	
Évora	30	0	0	
Lisboa	30	3	10	
Leiria	35	1	2.86	

**Table 2 pathogens-11-01242-t002:** Epidemiological features of the wild cervids with a PCR-positive result to *B. coli*.

Sample ID	Host Species	Age	Sex	Location
C24	*Cervus elaphus*	Juvenile	Female	Coimbra
C39	*Cervus elaphus*	Adult	Male	Leiria
C68	*Cervus elaphus*	Adult	Female	Portalegre
C120	*Dama dama*	Adult	Male	Lisbon
C140	*Dama dama*	Adult	Female	Lisbon
C143	*Cervus elaphus*	Adult	Male	Lisbon

**Table 3 pathogens-11-01242-t003:** *Balantioides coli* sequences generated in the present study compared with sequences retrieved from GenBank showing higher identity percentages in BLAST analyses.

Sequence	Host Species	Variant	GenBank ID	Identity (%)	Reference Sequence
C24	*Cervus elaphus*	B	OM349058	99.5	MT258438
C39	*Cervus elaphus*	A	OM349059	86.7	MZ676835
C68	*Cervus elaphus*	A	OM349060	88.2	JQ073346
C120	*D* *ama dama*	B	OM349062	83.0	MT252079
C140	*D* *ama dama*	B	OM349063	99.7	MT258433
C143	*Cervus elaphus*	B	OM349064	98.9	MT258438

## Data Availability

Not applicable.

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
