# Peer review of "Balantioides coli Fecal Excretion in Hunted Wild Cervids (Cervus elaphus and Dama dama) from Portugal"

_pathogens, 2022, doi:10.3390/pathogens11111242_

Round 1
Reviewer 1 Report
It is a well-written article that presents interesting results and considerations on “Balantioides coli fecal excretion in hunted wild cervids (Cervus elaphus and Dama dama) from Portugal” which is relevant due to the scarce literature on this topic.
Please consider some minor comments to the manuscript, as follow:
Line 68 – Replace M. tuberculosis by M. bovis
Line 73 – Remove from the aims of this study: to assess zoonotic potential and public health significance of B coli in wild cervids living in Portugal.
Material and methods – Add epidemiological information of interest regarding hunting areas where the samples were collected: Type of hunting area, Fenced/Open, Animals densities, Presence of wild boar, Evisceration on the field…
Results and discussion – Introduce the aforementioned topics in these sections.
Line 183 – Replace “slaughtered animals” by “hunted animals”.
Reviewer 2 Report
Balantioides coli is a zoonotic enteric protozoan parasite, many animals including cervids can be infected with this parasite and become its natural reservoirs for human balantiosis. This study use molecular tool to investigate the prevalence of B. coli in red deer and fallow deer from Portugal. The data showed that the positive rates of B. coli are 4.2% and 5.7% in red deer and fallow deer, respectively. The authors identified B. coli genetic variant A and B in the investigated deer. These findings will provide some important data for epidemiology of B. coli infections and be helpful for understanding the public health of this pathogen.
However, there are some problems in the manuscript, and need to be modified.
1. For statistical analysis, the names and versions of softwares or tools should be shown in the materials and methods section.
2. It is recommended to supplement the sequence similarity analysis within 6 positive samples investigated in this study to describe the sequence differences.
3. The results showed that low occurrence of B. col in both red and fallow deer. So some related descriptions are not reasonable in the text, and should be corrected. For example, Line 125-126: “Our results show that B. coli circulates widely among wild cervids in Portugal”, the word “widely” should be deleted; Line 190-191: Our study shows that a considerable number of wild deer in Portugal are infected with B. coli. The word “considerable” is used incorrectly here, and should be modified.
